# One pattern to express them all? Towards Generalised Patterns for Ontology Design in the Digital Humanities

Sarah Rebecca Ondraszek[1,*], Bruno Sartini[2], Marieke van Erp[3], Andrea Poltronieri[4], Pasquale Lisena[5] and Harald Sack[1,6]

[1]*FIZ Karlsruhe – Leibniz Institute for Information Infrastructure, Eggenstein-Leopoldshafen, Germany*
[2]*Ludwig-Maximilians-Universität München, Germany*
[3]*KNAW Humanities Cluster, Amsterdam, Netherlands*
[4]*Department of Computer Science and Engineering, University of Bologna, Italy*
[5]*EURECOM, Sophia Antipolis, France*
[6]*Institute of Applied Informatics and Formal Description Methods (AIFB) of KIT, Karlsruhe, Germany*

## Abstract

The digital humanities is a diverse area of research, encompassing just as diverse data as disciplines, thus posing particular challenges for ontology engineering. This paper pushes the application of ontology design patterns to solve these recurring challenges. It proposes a set of potential generalised patterns identified in various DH-related case studies, which address recurring modelling issues in the DH domain: annotations, knowledge provenance, or the interpretation of cultural artefacts. Functioning as a proof-of-concept, the work showcases how patterns could improve interoperability and knowledge capturing in ontologies for the DH.

## Keywords

Digital Humanities, Ontology Design Patterns, Semantic Technologies, Ontology Engineering

## 1. Introduction

Digital humanities (DH) is a diverse research field, encompassing (art) history, musicology, literary studies and more. To make sense of the equally diverse data sources, research questions and methodologies in this field, interdisciplinary projects require robust knowledge representations. The reuse of existing, widely accepted ontologies like the CIDOC Conceptual Reference Model (CIDOC CRM) [1] or Functional Requirements for Bibliographic Records (FRBR) [2] is already common practice. However, the unique nature of data resources in the DH often requires more specialised concepts. Significant challenges remain, such as guaranteeing consistent, comprehensible ontological representations. The comprehension and ability to model special

*15th Workshop on Ontology Design and Patterns (WOP 2024), ISWC 2024, November 12, 2024, Baltimore, MD*
*Corresponding author.

✉ sarah-rebecca.ondraszek@fiz-karlsruhe.de (S. R. Ondraszek); b.sartini@lmu.de (B. Sartini); marieke.van.erp@dh.huc.knaw.nl (M. v. Erp); andrea.poltronieri2@unibo.it (A. Poltronieri); pasquale.lisena@eurecom.fr (P. Lisena); harald.sack@fiz-karlsruhe.de (H. Sack)

🆔 0009-0003-7945-6704 (S. R. Ondraszek); 0000-0002-9152-4402 (B. Sartini); 0000-0001-9195-8203 (M. v. Erp); 0000-0003-3848-7574 (A. Poltronieri); 0000-0003-3094-5585 (P. Lisena); 0000-0001-7069-9804 (H. Sack)

situations and reuse useful pieces in the aforementioned standard ontologies can be challenging, pushing researchers to generate over-specified extensions or entirely new, domain-specific ontologies from scratch to represent knowledge in detail [3, 4]. While this might enable precise notions about the domain, the solutions often lack interoperability and reusability [5, 6]. Extending already complex models can foster equally complex queries, making them hard to use for non-experts. The same applies to ontology mappings, where such models require a thorough understanding of the underlying schema [7].

This paper explores a pattern-based approach to improve ontology engineering processes in the DH. Starting from the exploration of Ontology Design Patterns (ODPs) in given case studies, it promotes the abstraction of generalised patterns, assuming that there are problems specific to the DH domain that could be overcome with common, overarching solutions provided by modules with explicit documentation of underlying design principles [4].

In particular, we aim to address the need to adapt complex data models to specific use cases, emphasising the use of flexible and modular ontology components to improve interoperability and reusability in DH projects. This approach strives to foster enhanced consistency and greater coherence in ontology development practices. Instead of working top-down with a complex, existing ontology, the paper suggests starting with abstract patterns and specialising them in order to map them back to the appropriate best practice ontologies. We do this by discussing four case studies: 1) ICON [8, 9], 2) Polifonia Ontology Network [10], 3) Odeuropa [11], and 4) Viewsari [12], where we outline the utility of patterns for ontology design in the DH domain considering the heterogeneity of data sources (textual documents, visual artefacts, or intangible entities). Across the use cases, we analyse the patterns developed to identify commonalities and differences from which we extract patterns that could address multiple use cases from the domain. Our main contribution is a first resolution of the disparity between domain-specific ontologies from our case studies, showing that despite their differences, the patterns and the underlying traits could be shared across projects.

The remainder of this paper is organised as follows. Section 2 outlines the related work. Section 3 introduces the four case studies, followed by Section 4, which presents the comparative analysis between them. Finally, the discussion and conclusion in Section 5 summarise the paper.

## 2. Related Work

In this section, we outline best practices in ontological engineering, focusing on ODPs, their benefits, and their application in the DH domain.

Design strategies guide the creation of ontologies [13]. ODPs are modelling solutions for ontology design and address recurring design problems. They can come in various shapes, although for this work, only content patterns are of interest. They address content problems, providing solutions for domain modelling issues via encoding conceptualisations specific to domain classes and properties. These patterns enable a modularisation of the proposed frameworks, keeping them small and sectioned for use cases. As smaller frameworks – providing smaller windows through which the ontology can be viewed – they can reduce the complexity, making it easier to understand, maintain, query and reason (about) them [4].

In the domain of DH, modules and patterns have been applied in various cultural heritage

projects, some presented in this work as case studies. Another example is the ArCo KG [14], which integrates resources from the Italian cultural heritage.

Accurate knowledge representations in DH potentially require many diverse ontologies to cover the heterogeneity of the data sources [15]. Although some projects create domain-specific models for this, many rely on existing, often complex standards, such as the aforementioned CIDOC-CRM or FRBR. However, there is no standard for connecting all these ontologies [7]. Endeavours aiming to extract representative patterns from such core ontologies exist, e.g. the extraction of a pattern for Recurrent Events from CIDOC [16], or the attempt at the modularisation of CIDOC as a whole [17].

## 3. Case Studies

In this section, we describe our four case studies. These were selected based on a) their relevance to ontology development and the digital humanities domain, and b) their coverage of a wide range of domains and associated data resources.

### 3.1. ICON

ICON is an ontology that formalises the act of an interpretation following the theory of Panofsky [18], which divides artistic interpretations into three levels: pre-iconographic, iconographic, and iconological. At the pre-iconographic level, the interpreter recognises common motifs such as expressions, actions, and natural elements (i.e., animals and plants). The iconographic level identifies specific subjects (i.e., characters, symbols, personifications). At the iconological level, the interpreter recognises in the artwork deeper meanings related to cultural phenomena of the time of the creation or specific themes associated with the artist. ICON was developed to fill the gaps in the descriptions of iconographic and iconological statements in knowledge graphs [19] and was used as the schema of artistic knowledge graphs such as IICONGRAPH [20]. Following interoperability and reusability principles, ICON is aligned with and reuses several ontologies, including the Simulation Ontology [21], CiTo [22], HiCo [23], ArCo ontology [14].[1] ICON also reuses various ODPs.

The ontology distinguishes between recognitions (single acts) and interpretations (documents that can be supported/satisfied by multiple recognitions). To translate this distinction into the ontological model, the `Situation` and `Description` patterns from DOLCE Ultralight [24, 25] are reused. Each recognition is modelled following the `dul:Situation` pattern, which includes an agent performing it. The interpretation instead follows the `dul:Description` pattern. In DOLCE, descriptions are linked to situations via the `dul:isSatisfiedBy` property. Correspondingly, ICON specialises this relationship by three different properties according to the level of interpretation. Figure 1 shows the specialisation of the Situation and Description patterns in the ICON ontology.

Apart from explicitly mentioned patterns, two more emerge from the structure of ICON. First, at the pre-iconographic and iconographic levels, such a pattern highlights a relationship between the artwork, what the interpreter believes to be a specific depiction of something (in

---

[1]For a detailed description of the ICON ontology, we refer to its documentation https://w3id.org/icon/docs/.

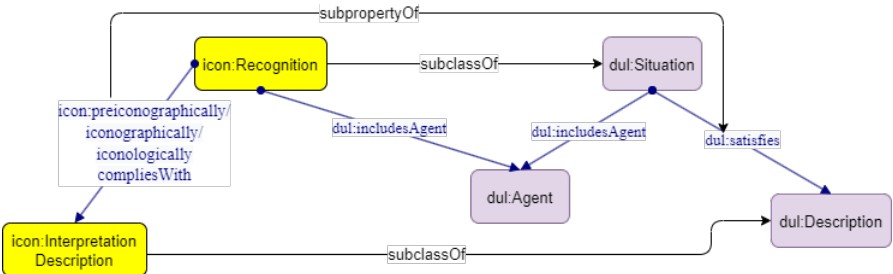

**Figure 1:** Situation and Description Patterns Specialisation in ICON.

the artwork), and the "universal" concept of what is depicted. For instance, in the Last Supper by Leonardo Da Vinci, the figure next to the right art of Jesus can be recognised as a depiction of a man. This recognition (`PreiconographicalRecognition`), in ICON, involves the artwork (ICON class: `Artwork`), the interpreter (`Agent`), the portion of the artwork that the interpreter associates with a depiction of a certain concept (`ArtisticMotif`), and the universal concept that the interpreter links with the depiction (`NaturalElement`). This pattern repeats on the iconographical level (`IconographicalRecognition`), where an interpreter recognises in the artwork the depiction of a specific subject (`Image`) and then links this depiction to the universal concept of that subject (like a `Character`). The emerged pattern is similar to the Semiotics pattern[2] with the difference that in the pattern the whole interpretation of meaning is based on linguistics, whereas ICON deals with visual and artistic meanings and there is an ongoing debate as to whether art can be considered part of linguistics [26]. Another pattern that emerges is the influence of one recognition on other recognitions of the same artwork. In the same example of the Last Supper, recognising a man in the artwork then influences the iconographical recognition of St. John, supporting the deeper meaning of the Artwork of a traditional representation of the Bible. However, other scholars suggested that the artwork depicts a woman, which then changes the iconographical recognition to Mary Magdalene and the overall deeper meaning deviates to a heretical reconstruction of the biblical scene, highlighting the close relationship between Jesus and Mary Magdalene [27].

### 3.2. Polifonia

Funded by the EU Horizon 2020 Programme, the Polifonia project[3] (2021-2024) aimed to recreate and make publicly accessible the connections between music, people, places, and events from the sixteenth century to the modern day through a comprehensive knowledge graph. A central contribution of the Polifonia project is the development of the Polifonia Ontology Network (PON) [10], comprising four main modules: Music Meta, Representation, Source, and Instrument.

PON features a pattern-based core design exemplified by the Music Meta module. Existing ontologies, such as the Music Ontology (MO) [28] and the DOREMUS ontology [29], while effective within their specific musical domains, encounter limitations when applied more broadly due to their genre and style-specific focuses and the diverse needs of various stakeholders.

---

[2]http://www.ontologydesignpatterns.org/cp/owl/semiotics.owl
[3]https://polifonia-project.eu/

These constraints necessitate a flexible model capable of abstracting to a general level ("zooming out") while also providing detailed, domain-specific applications ("zooming in"), thus ensuring interoperability alongside the capacity to articulate complex music metadata effectively.

Central to the Music Meta module within the Polifonia project is the `Information Realisation` ODP, which distinguishes between 'information objects'—non-physical social constructs that carry information—and their 'realisations,' the physical manifestations of these information objects. This pattern allows for a clear demarcation between the content of a piece (e.g., a musical composition) and its various materialisations (e.g., performances), treating both as distinct types of information entities [30].

In contrast, the Music Ontology and DOREMUS employ different adaptations of the FRBR model—FRBRer and FRBRoo, respectively—which conceptualise bibliographic resources through four layers: Work, Expression, Manifestation, and Item. However, within the IR pattern, only the Expression and Item layers are used, with Work and Manifestation layers criticised for their non-informative and overly complex conceptualisations in the context of music [30]. For instance, FRBR's Work layer does not adequately accommodate improvisations or traditional music, which often lack a formal composition process [2]. Moreover, the Manifestation layer, while clear in bibliographic contexts (e.g., a printed book), introduces ambiguity and complexity when applied to music, where it could signify a recording, a score, a compact disc, or all these forms.

### 3.3. Odeuropa

The Odeuropa project (2021-2023) was a European H2020-funded research project aimed at capturing and investigating smell history and heritage.[4] In the project, chemists, historians, museologists, computer vision specialists, language technologists and semantic web researchers worked together to capture information about smell heritage – with a particular focus on olfactory experiences – and its impact from their different research perspectives. The data types processed in the project ranged from text and images to user surveys and chemical analyses. A core challenge here was to design a data model for all this information: information about the smell itself, but also its creation and perception [11].

The Odeuropa data model was designed based on 74 competency questions in seven categories created by domain experts from the project team. The questions revolved around things such as smells, practices, gestures and emotions. To optimise integration with existing LOD datasets, existing ontologies such as CIDOC CRM [1] were reused to express the core concepts in the domain. Additional information was encoded using, among others, PROV-O [31] and Friend of a Friend (FOAF) [32].

Following the common practice of CIDOC CRM, the information is organised as a set of interconnected events revolving around the *Smell*. The latter represents an olfactory entity that exists at a precise time and space, unique and non-repeatable, rather than a generalisation of similar but still different smells, as we can interpret the *smell of roses* or the *smell of tobacco*. The core of the ontology – depicted in Figure 2 – requires two events directly connected to the Smell: a `Smell Emission` and an `Olfactory Experience`. These are linked respectively to objective

---

[4]https://odeuropa.eu

(smell source, odour carrier, place, etc.) and subjective characteristics (quality, emotion, gestures, etc.), most of which are intended to be populated with concepts from controlled vocabularies structured as a schema. It is important to note that the primary information is not directly linked to the smell entity, but the two events: this choice derives from considering the challenge related to modelling an immaterial element such as the smell, and to the necessity of putting spatio-temporal information at the centre of the ontology. Each smell can potentially be linked to several olfactory experiences made by different people or at different times, each involving a different set of subjective assessments (involved emotion, expressed description); this allows to present multiple points of view on the same smell.

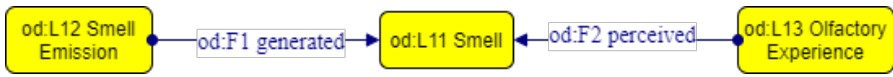

**Figure 2:** The Core Pattern of the Odeuropa Ontology.

In the resulting European Olfactory Knowledge Graph (EOKG) the 3 elements of the core are linked back to the original textual or image fragment using the PROV-O ontology; this allows one to always connect a smell experience with the metadata of the work from which it has been extracted.

### 3.4. Viewsari

Viewsari is a KG for Giorgio Vasari's Renaissance literary work, *Lives of the Most Eminent Painters, Sculptors, and Architects (The Lives)* [33]. It is a collection of biographical descriptions of prominent Renaissance artists, including detailed descriptions of their lives, works, and artistic styles. Being one of the foundational works of art history as a discipline, it remains a valuable resource for scholars and anyone interested in the period [34].

To describe contextual implications regarding relationships between entities, Viewsari distinguishes three layers: the bibliographic, the structural, and the content component. The bibliographic layer represents information about the work and its different analogue or digital manifestations. The structural layer contains information at the document level. In the third layer, the ontology encodes extracted content (named entities, co-occurrences). They are linked to their provenance information (e.g. paragraphs in which a co-occurrence occurs), represented in the structural layer. The latter links back to the appropriate bibliographic components, allowing provenance information to be traced through detailed location identifiers (paragraphs and positional arguments in the text) and information about the manifestation(s) used. For interoperability, the first and second layers are mapped to FaBiO and DoCo, while the content component provides mappings to PROV-O and the Web Annotation Ontology.[567] Aiming to complement existing approaches for extracting biographical information and resulting social networks based on event-centred models [35], this also represents the underlying process of opening up the data via agency and process orientation.

---

[5]https://sparontologies.github.io/fabio/current/fabio.html
[6]https://sparontologies.github.io/doco/current/doco.html
[7]https://www.w3.org/TR/annotation-vocab/#web-annotation-ontology

At the centre of Viewsari's knowledge representation is the co-occurrence of entities, shown in Figure 3. Concerning existing ODPs and approaches in other projects, e.g. the `Participation` pattern used in the MEETUPs ontology by Tirado et al. [36], a co-occurrence is a class, allowing for encoding of specific semantic meaning and annotation of complex information (e.g. additional attributes, provenance, or context).[8] For this reason, we refer to this pattern by `Annotation`. Similarly, Viewsari re-uses the `Provenance` pattern for traceability of statements for their source, source evidence for extracted entities, and co-occurrences in particular.[9]

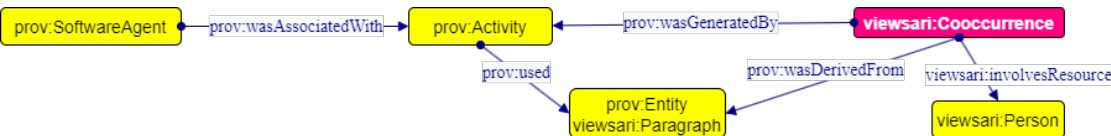

**Figure 3:** The Annotation of Co-occurrences in Viewsari.

Viewsari also reuses other existing patterns, including `Persons` and `Place`. While persons describe natural or social persons, the place pattern provides a structure for statements about the locations of things.[10][11]

Based on the multidimensional nature of Vasari's work, represented in Viewsari in the form of different layers, and the complex interplay between extracted content and contextual information, it resembles the nature of many DH projects. Corresponding with the previously outlined ontology, Viewsari comprises four core patterns relevant to the DH domain that could be further generalised. Next steps include extending the ontology with further annotations, including different perspectives from external sources, such as scholarly editions of *The Lives*. There are also plans to recognise artworks from mere textual descriptions in the work, suggesting the need to encode visual semiotics and descriptions in the model.

## 4. Comparative Analysis: Patterns for All!

In the following section, we undertake a comparative analysis of the four introduced case studies. Despite their differences in terms of general goals, data sources, or scope, we aim to identify common themes among the used or proposed patterns, bridging the gap between uniqueness and commonalities within the ontologies. A first methodological similarity concerns several included patterns that were extracted from or mapped to existing ontologies, e.g. DOLCE or CIDOC-CRM.

When directly comparing the patterns in Table 1, a fixed set of ten categories manifests: ICON's focus on formalising art interpretation is embodied in the Situation (SIT) and Description (DESC) patterns that help distinguish between recognition and interpretation. In the (pre-)iconographic levels, the Visual Semiotics (VIS-SEM) and Interpretation (INT) come into play, giving a logical backbone to the relationships between artworks, concepts, and their

[8]http://ontologydesignpatterns.org/wiki/Submissions:Participation

[9]http://ontologydesignpatterns.org/wiki/Submissions:Provenance

[10]http://ontologydesignpatterns.org/wiki/Submissions:Persons

[11]http://ontologydesignpatterns.org/wiki/Submissions:Place

**Table 1**

Comparisons of the adoptions of the different patterns of the case studies. Abbreviations: Situation (SIT), Description (DESC), Visual Semiotics (VIS-SEM), Interpretation (INT), Information Realisation (IR), Provenance (PROV), Person (PER), Place (PL), Annotation (ANN), Analogue vs Digital (AN-DIG)

| Case Study | SIT | DESC | VIS-SEM | INT | IR | PROV | PER | PL | ANN | AN-DIG |
|---|---|---|---|---|---|---|---|---|---|---|
| ICON | V | V | V | V | X | X | X | X | X | X |
| Polifonia | X | X | X | X | V | X | X | X | X | X |
| Odeuropa | X | X | X | X | X | V | X | X | X | X |
| Viewsari | X | X | X | X | X | V | V | V | V | V |

depictions. In particular, they show a direct link to how one recognition can influence other recognitions and subsequent interpretations. For Polifonia, the central pattern is Information Realisation (IR), allowing for the distinction between 'information objects' and their 'realisations', aiming to overcome challenges associated with FRBR's conceptualisation of resources. Odeuropa, while not explicitly involving patterns in the development process, still maintains patterns based on an event-centric model extending CIDOC-CRM on the one hand, and provenance information (PROV) encoded with PROV-O on the other. Viewsari features the Annotation (ANN) pattern for annotating co-occurrences. It also reuses the Person (PER) and Place (PL) patterns and represents manifestations of a literary artefact in the Analogue vs Digital pattern (AN-DIG). Akin to the use of PROV-O in Odeuropa, the PROV pattern enables the traceability of source information for extracted entities.

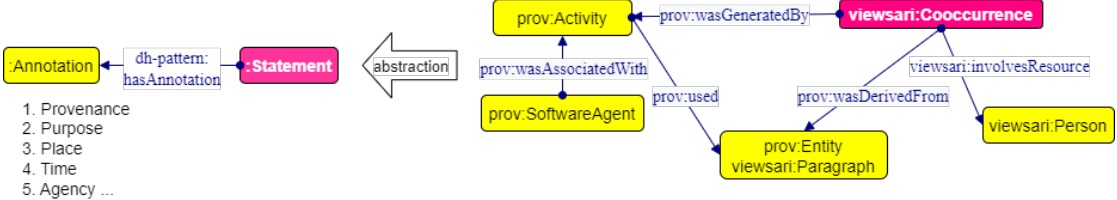

**Figure 4:** The Annotation of Statements, Abstracted from Viewsari's Annotation Pattern.

PROV is the only explicitly shared pattern across all our case studies. However, an extensive examination allowed us to identify several implicitly shared patterns, united by common features and characteristics. A first implicit overlap is the SIT, DESC, and VIS-SEM patterns, which help to articulate the nuances of artistic interpretation, scholarly analysis, and the representation of how artworks and interpretations are related. This is crucial for art historians, potentially benefiting Viewsari's ontology by encoding the evolution of textual interpretations.

Although not explicit in ICON, similar to Polifonia, there is a distinction between abstract concepts of an object and its manifestation. While Polifonia uses the IR pattern to encode different variants of a musical artefact, ICON focuses on the levels and states of (intangible) interpretation. This pattern could be useful in ICON to further separate the conceptual entity

depicted in artworks from the artworks themselves, not only via ad hoc created classes but also with the reuse of a foundational ontology. Similarly, in Viewsari, representations of the source material as digital or analogue artefacts (AN-DIG pattern) conceptualise the manifestation history, representing the immaterial, intellectual elements associated with Vasari's work. In addition, the extracted entities and their relations (co-occurrences) form abstract concepts for representing the content of said work. In Odeuropa, the structure of the odour emission, the immaterial odour itself and the experience represent a form of IR.

The PROV pattern is applied in Odeuropa and Viewsari. In both ontologies, it is a core structure to track the origins and contexts of entities, linking them to historical sources. Potentially, this could also be used to link interpretations in ICON to a certain source. For Polifonia, PROV can help trace the origins of musical works and their influences, tracking encounters of different musicians [36].

While only Viewsari reuses the PER and PL patterns, their generic nature makes them relevant to other domains. For example, they could be used to categorise historical figures and locations in Odeuropa, further enriching the context of olfactory heritage, given that it already uses FOAF to do so.

To the annotations, the same applies. While the annotation of co-occurrences with context information and provenance in Viewsari is – to some extent – domain-specific, its principles can be transferred to the other case studies, e.g. an act of interpretation in ICON can be annotated with more meta-information. Figure 4 shows the conceptualisation behind this. Accordingly, the ANN pattern in Viewsari complements / abstracts event-based approaches. It can help to provide scientific context and detailed analysis of data, linking it to historical texts and interpretations. Polifonia could reuse it to link compositions and performances to critical reviews, historical contexts and scholarly analysis. At the same time, both annotations in general and interpretive acts in ICON could be understood as events in the sense that they are contextual: they involve agents, their subjective evaluations when engaging with cultural artefacts, and related temporal elements.

The comparative analysis showed that there are few explicit commonalities between our selected case studies. Their patterns aim at handling textual, musical, sensory and visual data - thus they remain domain-specific, suggesting the need for tailor-made solutions in the field of DH. Despite these considerations and the fact that not all ontologies use the same patterns, common features could be identified that could further optimise ontology design in DH, suggesting common implicit patterns. The broad applicability of patterns such as AN and INT suggests relevance in terms of addressing common features of humanities data, potentially making them generalised patterns powerful enough to express a variety of use cases.

## 5. Discussion and Conclusion

This paper investigates how complex data models can be adapted to specific use cases in DH. It focuses on a pattern-based approach, investigating how patterns created in existing projects – such as Viewsari's annotation pattern – can be extended into generalisations suitable for DH-relevant problems.

To this end, the methodology involves a comparative analysis between the case studies ICON,

Odeuropa, Polifonia and Viewsari and their developed patterns to identify their similarities and differences. Then, through explicit or implicit overlapping of features from the projects and their knowledge representations, several generalised, common patterns could be identified, and their effectiveness in terms of addressing common features of humanities data assessed. By bridging the unique features and commonalities of all these applications, the patterns allow domain-specific goals to be aligned with more general concepts shared in DH.

Despite the reuse of patterns in the given case studies, only the PROV pattern is shared by several, suggesting that modelling in DH projects is still not fully based on standardised modules. Both Odeuropa and Viewsari use it to trace the origins of entities extracted from given sources, confirming the importance of provenance information in humanities research. Viewsari's use of additional patterns such as PER, PL, ANN and AN-DIG shows how patterns can be reused in detailed knowledge representation.

The intended use of patterns to enable simpler representations of complex content is exemplified by the PROV and ANN patterns, which can be used as a starting point for modelling documentation and scholarly discourse, providing the detailed context and traceability of interpretations needed when dealing with (art) historical data. Depending on the context, they can be adapted to the specific requirements of the application. Furthermore, as patterns such as DESC, SIT or PROV have been extracted from best practice ontologies – representing core concepts from them – they can still be mapped back to them, providing interoperability.

Widespread adoption of these patterns would improve interoperability, consistency and the ability to share and integrate data across projects. Further, this way, it could be evaluated to what extent the proposed patterns are valid in other use cases.

In summary, the reuse of ODPs in these case studies underscores their value in organising and interpreting complex, multisensorial, and interdisciplinary humanities data. This paper shows the potential of these patterns to meet diverse scholarly needs. The reuse of patterns improves individual projects and fosters a more interconnected digital humanities field, demonstrating the significant impact of ODPs on the advancement and connection of interdisciplinary (DH) research. The next steps include the investigation of the patterns' performance, assessing how the resulting KGs perform on general-level federate queries.

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
