# OpenReview forum: "One pattern to express them all? Towards Generalised Patterns for Ontology Design in the Digital Humanities"
_swsa.semanticweb.org/ISWC/2024/Workshop/WOP — WOP 2024 Oral_

### Official Review · Reviewer_vRKR · 2024-09-02
**Interesting topic and approach, good analysis, results a bit disappointing**

**Rating:** 6
**Confidence:** 3

**Review:**

The paper is 10p + 36 references
The main objective is to investigate how ODPs can be created in existing DH project can extended so as to be suitable for DH-relevant problems in general.
The methodology consists in analysing the use of ODPs in the ontologies of four DH-related projects (ICON, Polifonia, Odeuropa, Viewsari), judged representative of the domain, and compare these ODPs.
The result is a little bit disappointing, with one of the main conclusions being that:

- most generic patterns (describe places, persons, provenance, annotations), may be useful for different ontologies
- the case studies have too little in common to motivate the sharing of patterns

The abstract and introduction are well written

The related work is quite short. The first part is really just preliminaries about ODPs, and the second paragraph cites four related work about ODPs + DHs (mostly CIDOC-CRM).
It would be appropriate to motivate the choice of the four projects in the case study.

The section 3 (5 pages) introduces the four projects and describes the ODPs they reuse. This section is very well written, but maybe provides too much details not essential to the paper itself.

The comparative analysis is in Section 4 (2.5 pages). Table 1 compares the adoption of the 10 identified patterns by the 4 case studies.
Apart from PROV (provenance information), each pattern is used by just one project.
The paper continues with a high-level discussion of the implicit overlap between patterns, how patterns may be interchanged, reused, or generalized.
One concrete proposal is the generalization of the Viewsari's annotation pattern to show how it could be applied in one other project (ICON). Yet, the resulting ODP is really generic (Statement has annotation Annotation)

The topic is interesting and the discussions provided in this paper is surely of interest to the community.
However my overall feeling is that this study is quite preliminary, and that it provides a negative result: "The case studies are too different, only very generic patterns could be reused across projects".
It may be interesting to see to what extend the four identified ontologies could be rewritten (with transformation rules) so as to conform to a set of common patterns

---

### Official Review · Reviewer_XaNr · 2024-09-02
**Good analyzis of patterns used in DH field, but needs stronger arguments and clarity**

**Rating:** 6
**Confidence:** 4

**Review:**

The paper "One Pattern to Address Them All? Towards Generalized Patterns for Ontology Design in Digital Humanities" addresses the fragmentation of ontological resources in the digital humanities (DH) field, which challenges interoperability, reuse, and seamless access to and querying of knowledge graphs. The paper provides an analysis of ontology design patterns used across different projects (four use cases), covering various sub-areas of the DH domain, and aims to identify common design solutions that can be adopted in future ontology design initiatives, avoiding overly specialized ontologies and from-scratch efforts that lead to low reusability.
The paper is generally well-written, providing a good analysis of the selected use cases, with detailed descriptions of their design choices. Its contribution provides value in supporting the adoption of design patterns in future DH ontology design projects, which aligns with the workshop's objectives.
However, the paper falls short in some areas:
- The PROV-O ontology is presented as a pattern (which is more considered as an ontology, this is not the main issue) and is analyzed inconsistently. The authors stress that it represents the only explicitly common pattern used across the four projects: "PROV is the only explicitly shared pattern across all our case studies." However, Table 1 shows no use of PROV-O in ICON and Polifonia. Later, it is suggested as a possible solution for provenance tracking in these two use cases. Can you clarify ?
- The Related Work section should introduce the ontologies and patterns used in the DH field. For the WOP audience, these details would be more relevant than a general recall of pattern definitions. The scope of CIDOC-CRM and FRBR, identified as reference ontologies, can also highlight the importance of the study.
- The methodology for selecting the use cases should be better motivated. A brief explanation of why these projects were selected and to what extent they cover a wide range of cases in the field is necessary to support the intended generalization of the identified patterns.
- Identified patterns are not that common, which raises questions about the fragmentation of sub-domains. only prov-o (general purpose ontology for provenance tracking) can be shared across all projects. What about other use cases not covered ?
- A minor comment regards the paper's title, which could be misleading. "one pattern for them all?" might suggest that a single pattern could be enough for diverse design approaches, which is not the paper's purpose. Identified patterns are not common to the four UCs and could be generalized across only some of them.

The paper presents a valuable analysis of common design considerations in the DH field and suggests tracks to reuse some identified content patterns, which has the potential to increase ODP adoption in the field. Clarifying the use of PROV-O and providing a better introduction to the field (and selection methodology) would help increase the paper's quality and impact.

---

### Official Review · Reviewer_vofy · 2024-09-04
**Interesting preliminary study to identify common patterns but limited scope**

**Rating:** 6
**Confidence:** 4

**Review:**

The paper introduces a pattern-based approach to enhance ontology engineering processes within the Digital Humanities (DH) domain. It emphasizes the use of Ontology Design Patterns (ODPs) to address domain-specific challenges. The paper explores this methodology through four case studies with diverse data sources —ICON, Polifonia, Odeuropa, and Viewsari. The main contribution of the paper is resolving disparities between domain-specific DH ontologies from the case studies, demonstrating that despite their differences, shared patterns and underlying traits can be applied to various DH projects.

**Strengths**
- An increasingly important topic to address as more DH ontologies are being published
- The work is in scope of the workshop
- The writing is a clear and includes multiple figures
- The comparison section detailing the different ontologies provides useful insights into common patterns used in DH ontologies
- Next steps are interesting. It would be useful to have results from the performance of queries on these data models to guide DH projects when developing ontologies

**Weakness**
- Related work is very short and lacks sufficient detail. I would expect a more detailed discussion on ODPs related to DH
- Selection criteria of case studies is weak. I would expect a discussion of the methodology used to decide which projects to compare and contrast
- Lacking detailed description (short mention of competency questions in 3.3) of ODPs used to formulate the ontologies involved in the case studies e.g. protege, competency questions etc. A more focused discussion on the methods used by the case studies would be beneficial. Also describing how the cases studies validated their ontology design would be useful for the readers
- Links in footnotes 8-11 are not working
- The conclusion of the work is lacking clarity due to some inconsistent statements  - “suggesting the need for tailor-made solutions in the field of DH.” ->  “This paper shows the potential of these patterns to meet diverse scholarly needs.”
- “showcases how patterns could improve interoperability and knowledge capturing in ontologies for the DH.” - evidence to support this statement is missing

Overall, I think this paper attempts to tackle an important issue and the work is within scope of the workshop. It is a step in the right direction, however, I feel that a more focused and detailed discussion of case studies, which were chosen using a methodology would strengthen the work.